# Atypical nucleus pulposus migration and calcification: A specific radiographic sign for lumbar rheumatoid spondylitis

**Haocheng Cui****, Jingming Wang, Lei Wang, Xiaoduo Xu, Weimin Huang***

Orthopedic Department, 960 Hospital of People's Liberation Army, Jinan, Shandong, People's Republic of China

* ever_23@163.com

## Abstract

### Objective

This study aimed to characterize a specific radiographic manifestation of lumbar rheumatoid spondylitis.

### Methods and materials

The patients diagnosed with lumbar rheumatoid spondylitis who underwent posterior lumbar fusion surgery between 1/6/2019 and 30/4/2023 in the Department of Orthopedic Surgery in our hospital were retrospectively studied (RA group). The patients diagnosed with lumbar disc herniation with nucleus pulposus migration (cranial or caudal) were also collected for comparison (control group). The clinical data and radiographic manifestations were compared and analyzed.

### Results

A total of 14 patients in the RA group and 36 patients in the control group were enrolled in the current study. In the RA group, seven patients had nucleus pulposus migration, among whom five patients exhibited distinct calcification (5/7, 71.4%). Of the 36 patients in the control group, 10 patients demonstrated migrated nucleus pulposus calcification (10/36, 27.8%). The migrated nucleus pulposus in the RA patients exhibited diffuse calcification, while the migrated nucleus pulposus in the control group exhibited spot-like or shell-like calcification. In addition, nucleus pulposus migration and calcification in the RA group were more likely to occur in older patients, affect the higher lumbar levels and combine with intervertebral space collapse compared with the control group.

**Data availability statement:** All relevant data are within the manuscript and its Supporting Information files.

**Funding:** The author(s) received no specific funding for this work.

**Competing interests:** The authors have declared that no competing interests exist.

## Conclusions

Nucleus pulposus migration and calcification is specific in RA patients. It may be a characteristic radiographic sign to establish a lumbar rheumatoid spondylitis diagnosis. The potential mechanism is still unclear and need to be further explored.

## Introduction

Rheumatoid arthritis (RA) is an autoimmune disease primarily involving multiple synovial joints, causing synovial inflammation, cartilage destruction, and bone erosion [1]. The most commonly affected sites in RA include proximal interphalangeal joint, metacarpophalangeal joint, knee joint, and hip joint [1]. The atlantoaxial joint has been extensively studied regarding spine involvement, and its pathological changes can result in atlantoaxial instability and neurological impairment in severe cases [2–4]. However, the relationship between RA and the lumbar spine has received limited investigation [5]. In fact, RA involvement in the lumbar spine may be significantly underestimated and often overlooked. Kawaguchi [6] reported that 40% of the patients with RA suffered from chronic low back pain. Sakai [7] found that 45.2% of the patients with RA for more than 10 years exhibited a lumbar lesion.

Recognizing rheumatoid spondylitis and distinguishing it from common lumbar degenerative diseases are of great clinical significance. A previous study demonstrated that patients with RA who underwent posterior lumbar fusion had a significantly higher incidence of postoperative sepsis, pneumonia, and thromboembolic events [8]. Patients with rheumatoid spondylitis also had higher rates of deep infection, implant failures [9], and adjacent segment diseases [10] during the mid-and long-term follow-ups.

Some lumbar radiographic manifestations have been reported as specific indicators of lumbar rheumatoid spondylitis. These include intervertebral space narrowing [7], endplate or facet joint erosion [11], and vertebral subluxation and osteoporosis [12]. Besides these radiographic signs, the present study found that patients with lumbar rheumatoid spondylitis often exhibited lumbar nucleus pulposus migration and calcification (NPMC). Unlike the calcification observed in intervertebral disc protrusions associated with lumbar degenerative diseases, this type of calcification exhibits a diffuse pattern. We refer to this condition as "atypical." We reported in this study the five cases of lumbar rheumatoid spondylitis with NPMC. No studies have been conducted on this imaging feature for lumbar rheumatoid spondylitis. Hence, this study aimed to describe this specific radiographic sign and characterize it by comparing it with lumbar degenerative diseases.

## Materials and methods

### Study design

The collection of pertinent data for comparative studies was initiated on 1/7/2023. The patients diagnosed with lumbar rheumatoid spondylitis who underwent posterior lumbar fusion surgery in the Department of Orthopedic Surgery in our hospital

between 1/6/2019 and 30/4/2023 were retrospectively studied (RA group). The patients diagnosed with lumbar disc herniation with nucleus pulposus migration (cranial or caudal) who underwent posterior lumbar surgery from 1/1/2021–30/4/2023 in the Department of Orthopedic Surgery in our hospital were also collected for comparison (control group). All the enrolled patients were operated by a single surgical team. The studies involving humans were approved by Institutional ethical committee of 960 hospital of PLA. The studies were conducted in accordance with the local legislation and institutional requirements. The participants provided their written informed consent to participate in this study. Written informed consent was obtained from the individual(s) for the publication of any potentially identifiable images or data included in this article.

## Inclusion and exclusion criterions

**Inclusion criteria.** The RA group: Patients diagnosed with RA according to the RA classification and diagnostic criteria established by American College of Rheumatology (ACR) and European League Against Rheumatism collaborative initiative (EULAR) in 2009 [13]; Patients who had lumbar involvement caused by RA (The involvement criterion in the current study was endplate or facet joint erosion); Patients suffered from low back pain and/or neurogenic lower extremity dysfunction; Patients who failed to conservative treatment and underwent posterior lumbar surgery.

The control group: Patients diagnosed with lumbar disc herniation with nucleus pulposus migration; Patients without rheumatic diseases such as RA, ankylosing spondylitis, psoriasis, systemic lupus erythematosus; Patients who failed to conservative treatment and underwent posterior lumbar surgery.

**Exclusion criteria.** Patients with tumor, trauma or infection in lumbar spine; Patients with previous lumbar surgery history; Patients with incomplete clinical data.

## Clinical assessment

The serum laboratory examinations including rheumatoid factor (RF), erythrocyte sedimentation rate (ESR), C-reaction protein (CRP) were retrospectively recorded. RF was assessed only in the RA group, as these markers are diagnostic criteria for RA and are irrelevant to degenerative disc disease.The rheumatoid progression and functional capacity were evaluated by the Steinbrocker classification [14].

## Radiographic assessment

All patients underwent lumbar plain radiographs, Computerized Tomography (CT) scans and Magnetic Resonance Imaging (MRI) examinations preoperatively. All images were reviewed independently by one specialist in lumbar surgery with more than 18 years of experience and one experienced radiologist with more than 16 years of experience.

**Plain radiographs.** The anterior-posterior and lateral lumbar radiographs were obtained in upright position. The spine endplate was assessed by the lateral plain radiographs. The endplate erosion degree was classified to mild (erosive), moderate (sclerotic) and severe (collapsed) [7]. The intervertebral space was documented as preserved and narrowing assessed by the lateral plain radiographs. Scoliosis was define as Cobb angle >10° on the anterior-posterior plain radiographs.

**CT scans.** The CT images were obtained using 64-slice CT (Sensation 64; Siemens, Erlangen, Germany) with patients in the supine position. Calcification was defined as hyperdense lesions exceeding 200 Hounsfield units on CT scans. Besides, the calcification features were described and compared between groups.

**MRI scans.** The MRI images of the lumbar spine were obtained using a 1.5-T scanner (SIGNA Creator; GE, Boston, USA). The MRI imaging protocols included axial and sagittal T2-weighted images, sagittal T2-weighted sequences with fat saturation, and sagittal T1-weighted images. In the sagittal MRI plane, migration was defined as cranial or caudal from the disc level upward or downward respectively. In the axial MRI scans, the facet joints erosion was graded as follows: Grade

0 was defined as no erosion; Grade 1 was defined as the eroded area occupied≤50% of the entire facet joints; Grade 2 was defined as the eroded area occupied≥50% of the entire facet joints [11].

## Statistical analysis

SPSS 14.0(SPSS version 14.0; SPSS Inc, IL) was applied for statistical analysis. The measurement data were described by the median (quartile), the Wilcoxon rank sum test was used for the comparison between the two groups, the counting data were described by the number of cases (percentage), and Fisher's exact probability method was used for the comparison between groups. P-values of less than 0.05 were considered to indicate statistical significance. The kappa values were calculated for intra-observer and inter-observer reliability between the two observers. The Landis and Koch interpretation of kappa values was used [15].

## Results

### Baseline characteristics

A total of 14 patients in the RA group and 36 patients in the control group were enrolled in the current studies. Baseline data of RA group were showed in Table 1.

**Radiographic outcomes.** The intraobserver and interobserver reliability both showed substantial agreement (k = 0.68 and k = 0.70). Of the 14 RA patients, all exhibited endplate destruction, of which two cases were mild (erosive) (14.3%), four cases were moderate (sclerotic) (28.6%), and eight cases were severe (collapsed) (57.1%). Regarding to vertebral space, one patient (7.1%) had normal vertebral space, the other 13 patients (92.9%) had vertebral space narrowing. On anteroposterior radiographs, 11 patients (78.6%) demonstrated scoliosis with the mean Cobb angle of 16°.

All patients underwent CT examinations. In the RA group, seven patients had nucleus pulposus migration, among whom five patients showed distinct calcification (5/7, 71.4%), while in the control group, 10 (10/36, 27.8%) patient showed calcification. All the five patients in the RA group exhibited atypical nucleus pulposus migration and calcification, while all the ten patients in the control group exhibited shell-like or spot-like calcification. The detailed information were demonstrated in Table 2 and Fig 1.

The MRI scans revealed that there were seven patients (7/14,50%) exhibitting nucleus pulposus migration, with four migrated cranially and three migrated caudally in the RA group. In the control group, there were 9 patients (9/36, 25.0%) with cranial migration and 27 patients (27/36, 75.0%) with caudal migration. The bilateral facet joints from L1/2 to L5/S1 in

**Table 1. Baseline characteristics of the enrolled cases.**

| Characteristics | RA | control |
|---|---|---|
| Age, years (average± SD) | 47-79 (62.9±9.9) | 30-86 (45.5±13.02) |
| Sex | | |
| Female | 10 | 15 |
| Male | 4 | 21 |
| Disease duration, years (average± SD) | 5-32 (17.9±9.6) | 0.02-2 (0.45±0.61) |
| Steibrocker's classification | | |
| Stage II | 1 | |
| Stage III | 9 | |
| Stage IV | 4 | |
| Class II | 4 | |
| Class III | 10 | |

**Table 2. Imaging characteristics of 5 RA patients with nucleus pulposus migration and calcification.**

| Number | Plain X-ray films | | | CT | MRI | |
|---|---|---|---|---|---|---|
| | Anteroposterior (Cobb angle) | lateral | | IVD | IVD | Facet joint[b] (Grade) |
| | | end plates[a] | IVD | | | |
| 1 | scoliosis (11°) | collapsed | narrowing | migrations, calcification | Caudal migrations (L3/4) | Grade0: L1/2, L4/5, L5/S1<br>Grade1: L2/3<br>Grade2: L3/4 |
| 2 | scoliosis (17°) | sclerotic | narrowing | migrations, calcification | Cranial migrations (L4/5) | Grade0: L1/2, L2/3, L5/S1<br>Grade1: L3/4<br>Grade2: L4/5 |
| 3 | scoliosis (14°) | sclerotic | narrowing | migrations, calcification | Cranial migrations (L3/4) | Grade0: L4/5, L2/3, L5/S1<br>Grade1: L1/2<br>Grade2: L3/4 |
| 4 | scoliosis (17°) | sclerotic | narrowing | migrations, calcification | Caudal migrations (L2/3) | Grade0: L1/2, L3/4, L5/S1, L4/5<br>Grade1: L2/3 |
| 5 | Normal (7°) | collapsed | narrowing | migrations, calcification | Cranial migrations (L4/5) | Grade0: L1/2, L2/3, L3/4, L5/S1<br>Grade1: L4/5 |

RA, Rheumatoid arthritis.

[a] The endplate erosion degrees were classified as mild (erosive), moderate (sclerotic), and severe (collapsed) [7].

[b] The facet joint erosion was graded as follows: grade 0 indicated no erosion; grade 1 as the eroded area occupying ≤50% of the entire facet joints; grade 2 as the eroded area occupying ≥50% of the entire facet joints [11].

the RA group were examined. All the patients in the RA group exhibited at least one level facet joint erosion, and of the 70 facet joints, there were 11 facet joints with Grade 1 (11/70, 15.7%) and 8 levels with Grade 2 (8/70, 11.4%).

When comparing the patients with nucleus pulposus migration and calcification between groups, it was detected that there were differences in mean age, involved levels, migration direction and intervertebral disc height. Except for migration direction, all the difference had statistical significances(Table 3).

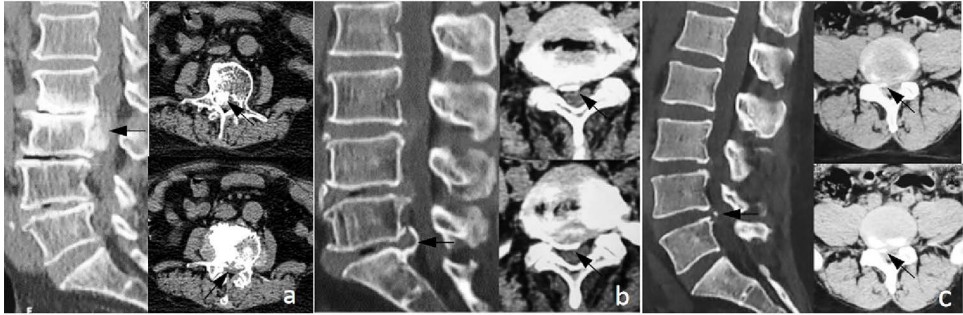

**Fig 1. Imaging findings of atypical and common disc calcification.** (a) Sagittal CT reconstruction showed the mass of the prolapsed intervertebral disc located in the spinal canal behind the L3 vertebral body, and the whole prolapsed intervertebral disc presented a diffuse high-density shadow. Furthermore, intervertebral disc was calcified, a large herniation occupying more than 1/4 of the intravertebral canal cross-sectional area (black arrow). (b, c) were common intervertebral disc herniation with calcification. (b) was marginal calcification, which was 'shell-like' (black arrow); (c) was marginal calcification, which was 'spotty' (black arrow).

 

**Table 3. Comparison of nucleus pulposus migration with calcification in RA patients and ordinary nucleus pulposus migration with calcification.**

|  | RA(n=5) | Normal(n=10) | P |
|---|---|---|---|
| Age(years), Median (IQR) | 50.0(48.5,59.0) | 35.5(29.5,48) | 0.013 |
| Gender |  |  | 0.329 |
| Male, n(%) | 2(40%) | 7(70%) |  |
| Intervertebral level, n(%) |  |  | 0.047 |
| L2/3 | 1(20%) | 0(0%) |  |
| L3/4 | 2(40%) | 0(0%) |  |
| L4/5 | 2(40%) | 8(80%) |  |
| L5/S1 | 0(0%) | 2(20%) |  |
| Migration in Vertical plane, n(%) |  |  | 0.077 |
| Caudal | 2(40%) | 9(90%) |  |
| Cranial | 3(60%) | 1(10%) |  |
| Intervertebral Disk Height, n(%) |  |  | 0.007 |
| Noramal | 0(0%) | 8(80%) |  |
| Narrowing | 5(100%) | 2(20%) |  |

## Case presentation

**Case 1.** A 51-year-old male with a RA history for more than five years was admitted in our department. He suffered from low back pain and lower limb pain for about six months. Antirheumatic drugs was not performed regularly. The laboratory examination showed an elevated ESR of 60 mm/h, CRP of 21.4 mg/L and RF of 481 IU/ml. Lateral radiographs displayed the L3/4 intervertebral space collapse, sclerosis of the L3 lower endplate and erosion of the L4 upper endplate. CT scans revealed massive calcified lesion on the dorsal side of the L4 middle part in the spinal canal. MRI demonstrated that nucleus pulposus migrated caudally in the spinal canal (Fig 2). Posterior decompression and fusion surgery at the

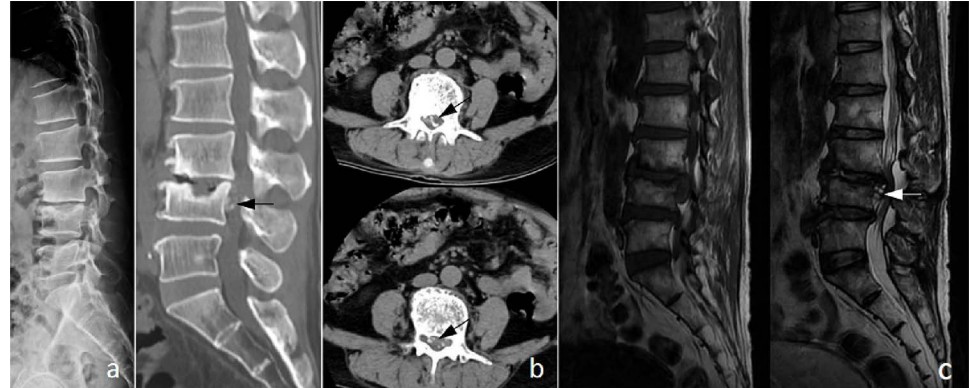

**Fig 2. Imaging manifestations of typical case 1: intervertebral disc protrusion with calcification.** (a) Lateral x-ray film of the lumbar spine showed significant narrowing of the L3/4 space, sclerosis of the L3 bottom endplate and depression of the L4 upper endplate. (b) Sagittal CT showed sclerosis of the L3 bottom endplate, obviously collapse of the L4 upper endplate and mass high-density shadow in the spinal canal behind L4 vertebral body (black arrow). CT scan showed scattered high-density calcification (black arrow) within the spinal canal, with a mass larger than 1/2 of the cross-sectional area of the spinal canal. (c) MRI examination showed that the L3/4 intervertebral disc was herniated and displaced downward, located in the spinal canal behind the L4 vertebral body, in clumps, with low T1 signal and mixed T2 signal (white arrow).

L3/4 level was performed. Preoperative complain was completely relieved without complication during the four years follow-ups.

Case 2. A 47-year-old male complained of low back pain and left lower limb pain for two years was admitted in our department for surgical treatment. He was diagnosed with RA five years ago with irregular oral methotrexate for about 9 months. The laboratory tests showed an ESR of 17 mm/H, CRP of 3.23 mg/L and RA of 397 IU/ml. The lateral radiographs and CT sagittal scans demonstrated that intervertebral space collapse and significant erosion of the L4 lower endplate and L5 upper endplate. The nucleus pulposus migrated cranially on the MRI scans and the migrated lesion exhibited extensively calcification on the CT scans (Fig 3). Posterior decompression and fusion was performed at the L3-L5 levels and the patient had a fast postoperative recovery and a good clinical outcome during the 42 months follow-up.

Case 3. A woman aged 50 had a history of seropositive RA for more than 30 years. She had oral methotrexate for about 11 years and changed to oral Corticosteroid in recent five years. She developed low back pain for five years and radiated down the left lower limb for one year. The laboratory tests showed an ESR of 16 mm/H, CRP of 20 mg/L and RF of 20.7 IU/ml. Lateral radiographs and sagittal CT scans demonstrated sclerosis of the lower L3 and upper L4 endplate, erosive destruction of the lower L3 endplate and narrowing of the L3/4 intervertebral space. The MRI scans detected nucleus pulposus migrated cranially with diffuse calcification showed in CT scans (Fig 4). Posterior decompression and intervertebral fusion was performed and the patient had a good recovery with occasional mild low back pain during the three years follow-up.

Case 4. A 67-year-old woman with RA history for more than 20 years had a complaint of low back pain radiating down the right lower limb pain for six months. Oral corticosteroid therapy was applied for three years. The laboratory tests showed an ESR of 10 mm/H, CRP = 3.23 mg/L and RF of 114 IU/ml. The lateral radiographs and CT scans showed the L2/3 and L3/4 intervertebral spaces were narrowing and sclerotic with endplate minor erosion. The L2/3 nucleus pulposus migrated down to the lower edge of L3 vertebral body on the MRI scans and was calcified diffusely on the CT scans. Postoperative histology examination indicated nucleus pulposus tissue with calcification (Fig 5). She was painfree postoperatively and had no complications during the 28 months follow-up.

Case 5. A 50-year-old female with a 15-year RA history developed low back pain and lower limb pain for five months. She had oral leflunomide and prednisolone for eight years. The laboratory tests showed an ESR of 14 mm/H, CRP of

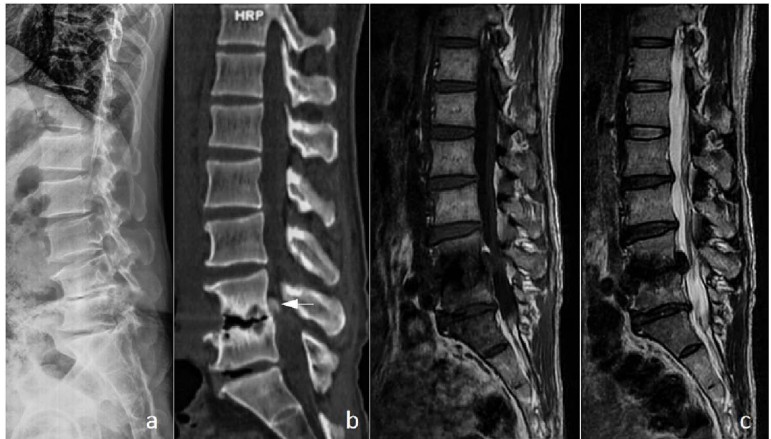

**Fig 3. Imaging manifestations of typical case 2: intervertebral disc protrusion with calcification.** (a) Lateral x-ray film of the lumbar spine showed significant narrowing of the L4/5 space and sclerosis of the L4 bottom endplate and L5 upper endplate. (b) Sagittal CT showed erosion and collapse of both L4 and L5 endplates, disappearance of vertebral space, and quasi-circular high-density shadow (indicated by white arrow) in the spinal canal behind L4 vertebral body. (c) MRI showed severe erosion of L4 and L5 endplates and L4/L5 disc prolapsed and upward displacement.

3.23 mg/L and RF of 15.4 IU/ml. The lateral radiographs demonstrated L4 spondylolisthesis with L4/5 intervertebral space collapse. The axial CT scan exhibited that a massive calcified lesion was located at the left lateral recess in the spinal canal. The MRI scans revealed the nucleus pulposus migrated cranially. Her symptom was successfully relieved after decompression and fusion surgery and no complication occurred during the 8 months follow-up (Fig 6).

## Discussion

The current study has demonstrated that nucleus pulposus migration and calcification is a specific radiographic sign for lumbar rheumatoid spondylitis. Compared with non-rheumatic patients, the migrated nucleus pulposus in the RA patients

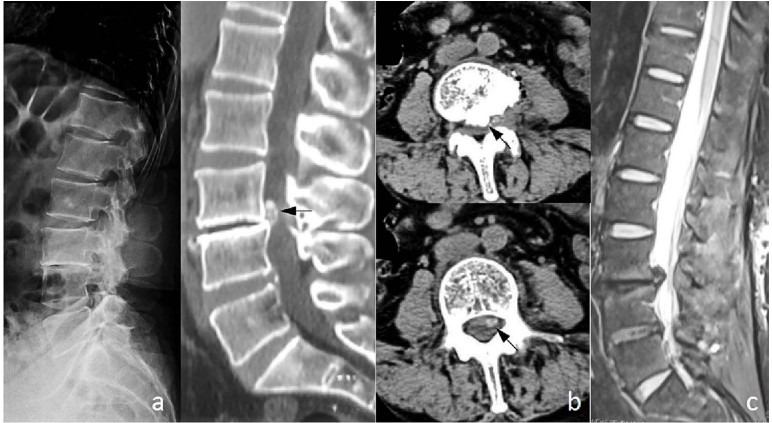

**Fig 4. Imaging manifestations of typical case 3: intervertebral disc protrusion with calcification.** (a) Lateral x-ray film of the lumbar spine showed sclerosis of the L3 and L4 endplate and narrowing of the intervertebral space. (b) Sagittal CT showed sclerosis of the L3 bottom endplate and L4 upper endplate and quasi-circular high-density shadow in the spinal canal behind L3 vertebral body. CT scan showed a prominent, high-density image in the spinal canal, with a left of center, occupying more than 1/2 of the cross-sectional area of the spinal canal (indicated by black arrow). (c) MRI showed slight erosion changes in L3 and L4 endplates, and L3/L4 disc prolapse and upward displacement.

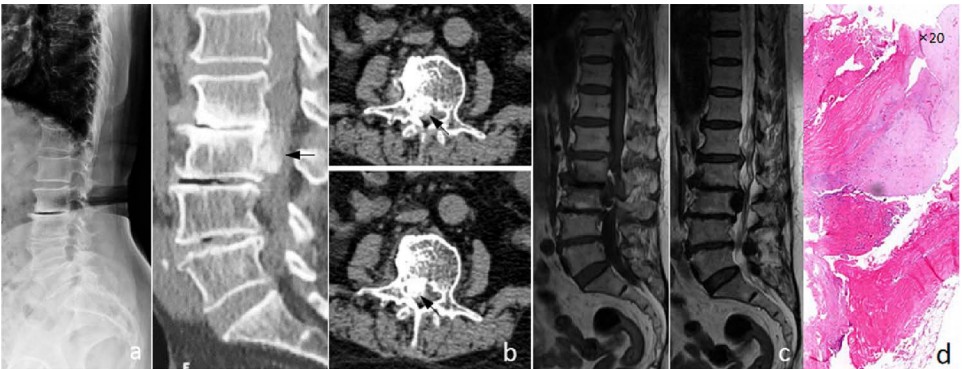

**Fig 5. Imaging manifestations of typical case 4: intervertebral disc protrusion with calcification.** (a) Lateral x-ray film of the lumbar spine revealed sclerosis of the L2 and L3 endplate and narrowing of the intervertebral space. (b) Sagittal CT showed sclerosis of the L2 and L3 endplate, massive high-density shadow (indicated by the black arrow) in the spinal canal behind the L3 vertebral body. CT scan showed occupying space larger than 1/2 of the vertebral canal cross-sectional area (indicated by black arrow). (c) MRI examination showed erosion changes in L2 and L3 endplates, narrowing of L2/L3 vertebral space, and L2/L3 disc prolapse and downward displacement.(d) Several pieces of broken nucleus pulposus tissue were removed from the spinal canal during the operation, and histology indicated that nucleus pulposus tissue with calcification.

exhibits diffuse calcification. Nucleus pulposus migration and calcification described in the current study is expected to be a novel radiographic feature to determine a lumbar rheumatoid spondylitis diagnosis.

With the increasing recognition on lumbar involvement in RA patients, some specific radiographic features for lumbar rheumatoid spondylitis have been reported. Lawrence first described the radiological features of lumbar lesions in RA patients, including intervertebral disc narrowing without osteophyte formation, lumbar spondylolisthesis, facet joint erosion, and osteoporosis [16]. Nakase et al. studied the changes of vertebral endplate in RA patients and proposed to divide the changes into four levels [17]. Sakai et al further improved Nakase's grading system. They reviewed 104 lumbar disease patients with at least 10 years of RA history and integrate the intervertebral space changes into the grading system [7]. Ohishi compared the progression of lumbar scoliosis in RA patients to non-RA patients and found that the clinical progression of scoliosis in RA patients differed from those in degenerative lumbar scoliosis [12].

So far, there has been no consensus diagnostic criteria on lumbar rheumatoid spondylitis. The current study established the lumbar rheumatoid spondylitis diagnosis by lumbar endplate or facet joint erosion which the authors considered as the most distinctive radiographic feature of lumbar rheumatoid spondylitis. Firstly, although spondylolisthesis, scoliosis and osteoporosis have been reported in RA patients in previous studies, they are not specific radiographic features for lumbar rheumatoid spondylitis. Spondylolisthesis and scoliosis are common radiographic findings in lumbar degenerative diseases and in RA patients, they could also be the consequences of facet joints and endplate destruction. Osteoporosis is a common clinical condition among the elderly women and it may also be the consequence of oral glucocorticoids in RA patients. Secondly, although RA is a systemic disease and a variety of extra-articular manifestations occur such as rheumatoid nodules, pulmonary involvement and vasculitis, synovitis is the best characterized pathogenesis [18]. The synovial tissue is transformed into a destructive pannus and osteoclasts are often observed in pannus invasion areas of bone which leads to cartilage destruction and bone erosion [19,20]. In lumbar spine, the facet joints have typical synovial joint structure and the vertebral endplate is similar to the synovial joints. Previous studies have also demonstrated that typically erosive destruction in the facet joints and the lumbar endplate in RA patients [21]. Therefore, the facet joints and endplate erosive destruction was taken as diagnostic criteria in the current study.

Nucleus pulposus migration and calcification is different from the intervertebral disc calcification (IDC) previously reported. IDC is not a rare radiographic manifestation and has been reported in various diseases, among which lumbar degenerative diseases are the most frequently reported. A number of studies have demonstrated that IDC have

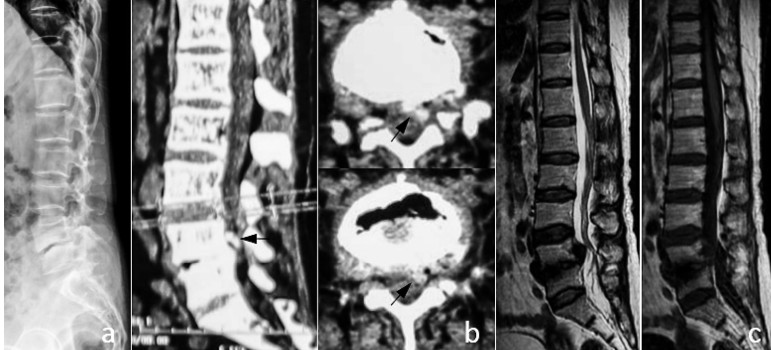

**Fig 6. Imaging manifestations of typical case 5: intervertebral disc protrusion with calcification.** (a) Lateral x-ray film of the lumbar spine revealed sclerosis of the L4 bottom endplate and narrowing of the intervertebral space. (b) Sagittal CT showed disappearance of the L4/L5 intervertebral space, massive high-density shadows were seen in the spinal canal behind the L4 vertebral body. CT scan showed high-density shadows were located to the left of the center of the spinal canal at the left lateral recess, occupying space larger than 1/2 of the vertebral canal cross-sectional area (indicated by black arrow). (c) MRI examination showed erosion of L4 and L5 endplates, narrowing of the intervertebral space, and L4/5 disc prolapse and upward displacement.

relationship with intervertebral disc degeneration [22,23]. Disc degeneration may promote the expression of transcription factors, bone morphogenetic protein 2 and osteocalcin, which might induce disc calcification [24–26]. IDC in lumbar degenerative diseases is different from that in RA. IDC in the degenerative diseases is often characterized by calcification in the marginal area of lumbar disc, which exhibits spot-like or shell-like calcification on the CT scans. In the current study, 10 patients in the control group have detected calcification and all of them have presented spot-like or shell-like calcification. This indicates the calcification in the lumbar degenerative diseases mainly deposits in the annulus fibrous tissue. In contrast, calcification of the migrated nucleus pulposus in RA patients have distinctive features that it exhibits a nodular lesion with diffused calcification. And this indicates the calcification in the lumbar rheumatoid spondylitis mainly occurs in the nucleus pulposus tissue.

IDC may also occur as pathologic changes secondary to some symptomatic diseases such as ochronosis, haemochromatosis, and hyperparathyroidism. In these conditions, IDC always involves multiple levels and is easy to differentiate from nucleus pulposus migration and calcification in RA patients. In some conditions, IDC conducted as the main pathological change. Pediatric idiopathic intervertebral disc calcification is a rare clinical condition of unknown etiology. It occurs predominately at the cervical and upper thoracic levels and has a favorable clinical outcome with a self-limited clinical course [27]. In the light of the onset age and involved levels, it is easy to distinguished from the nucleus pulposus migration and calcification in RA patients.

In contrast to pediatric idiopathic intervertebral disc calcification, rare information is available to adult idiopathic intervertebral disc calcification. Xu [28] reported that two adult male patients with sudden chest and back pain. CT sagittal reconstruction found that the calcified nucleus pulposus in the thoracic disc accompanied by huge calcified disc herniation into the spinal canal. The radiographic findings of the protruding calcified disc had similarities to those described in this paper, but were still different. The cases with adult idiopathic intervertebral disc calcification have the nucleus pulposus calcified in situ, while the cases with RA have the migrated nucleus pulposus calcified in the spinal cord. In addition, adult idiopathic intervertebral disc calcification occurred in thoracic spine, while the nucleus pulposus migration and calcification of RA all occurred in lumbar spine.

The relationship between RA and IDC was rarely reported previously. Nakamura et al [29] have reported one RA patient with multi-level thoracic and lumbar IDC. The MRI scans and lateral radiographic view showed the calcification was in situ and involved multiple intervertebral discs, which is significantly different from the nucleus pulposus migration and calcification documented in the current study.

The nucleus pulposus migration and calcification in lumbar rheumatoid spondylitis should also be differentiated from the rheumatoid nodules in the spinal canal. Hirohashi [30] reported a 51-year old woman a 9-year RA history developed symptomatic rheumatoid nodules in the lumbar epidural space. The MRI revealed that tumorous masses compressed the spinal sac at the L4/5 level resulting in bilateral L5 nerve root compression. Lumbar CT examination revealed a filling defect of L4/5 in the spinal canal with scattered calcification. Although rheumatoid nodules in the spinal canal and nucleus pulposus migration and calcification both exhibits nodular lesion with calcification, it is not difficult to distinguish the two conditions. Firstly, patients with rheumatoid nodules in the spinal canal presented with intact endplate and intervertebral disc space, while all the patients with nucleus pulposus migration and calcification presented with obvious endplate erosion and intervertebral disc collapse. Secondly, rheumatoid nodules was found to have a yellow membrane during the operation [30], while the migrated and calcified nucleus pulposus in the current series were found to have rough surface without membrane structure. Finally, the postoperative photomicrographs of rheumatoid nodules demonstrated extensive fibrinoid necrosis surrounded by poorly formed palisades of histiocytes and chronic inflammatory cells, which indicated typical rheumatoid nodules. However, the postoperative photomicrographs of lesion sections in the current studies demonstrated degenerated nucleus pulposus tissue with calcification.

The mechanism of nucleus pulposus migration and calcification in RA patients is not clear. In the current study, it is noted that all the five cases with nucleus pulposus presented with obvious endplate erosion and sclerosis. In view of this,

the authors speculate that the RA specific cartilage endplate destruction and bony endplate erosion may play an important role in nucleus pulposus migration and calcification. It is well established that RA leads to cartilage destructions and bone erosion in lumbar spine [6]. The lumbar cartilage endplate has an important role in nutrient supply and metabolism for the nucleus pulposus [31]. It is postulated that cartilage endplate destruction and the bony endplate erosion or sclerosis in RA patients might reduce or occlude the nutritional channels to nucleus pulposus and consequently lead to increased anaerobic metabolism and lower metabolic rates, and finally trigger the formation of nucleus pulposus calcification. However, the exact mechanism remains poorly understood and need to be further investigated.

The main limitation of the current study is its retrospective nature with a relatively small single-center cohort. And for this reason, the correlative factors of nucleus pulposus migration and calcification including baseline characteristics, Steinbrocker classification, other radiographic features of rheumatoid spondylitis were not included in the analysis of this study. Another limitation of the study was that all the enrolled patients had serious symptoms requiring surgical intervention, and this may lead to a sample-selection bias. Large-sample multicenter studies are warranted to further support our conclusions, explore the relative factors and observe the sensitivity and specificity in diagnosing rheumatoid spondylitis.

## Conclusions

NPMC is a specific feature in patients with RA. It may be a characteristic radiographic sign to establish a diagnosis of lumbar rheumatoid spondylitis. However, the potential mechanism is still unclear and needs further exploration.

## Supporting information

**S1 Dataset.**
(XLSX)

**S2 Dataset.**
(XLS)

## Author contributions

**Data curation:** Jingming Wang, Xiaoduo Xu.

**Formal analysis:** Lei Wang.

**Writing – original draft:** Haocheng Cui.

**Writing – review & editing:** Weimin Huang.

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
