## [Decision Letter · Decision Letter 0]

11 Mar 2025

PONE-D-24-49692Atypical Nucleus Pulposus Migration and Calcification: A Specific Radiographic Sign for Lumbar Rheumatoid SpondylitisPLOS ONE

Dear Dr. Cui,

Thank you for submitting your manuscript to PLOS ONE. After careful consideration, we feel that it has merit but does not fully meet PLOS ONE’s publication criteria as it currently stands. Therefore, we invite you to submit a revised version of the manuscript that addresses the points raised during the review process. The reviewers have highlighted some issues which can be solved but do need to be addressed in order tha the Manuscript can be considered adequate for publication.

We look forward to receiving your revised manuscript.

Kind regards,

Alessandra Aldieri

Academic Editor

PLOS ONE

Journal Requirements:

2. We note that your Data Availability Statement is currently as follows: “All relevant data are within the manuscript and in Supporting Information files.”

Reviewers' comments:

Reviewer's Responses to Questions

**Comments to the Author**

1. Is the manuscript technically sound, and do the data support the conclusions?

Reviewer #1: Yes

Reviewer #2: No

2. Has the statistical analysis been performed appropriately and rigorously? 

Reviewer #1: Yes

Reviewer #2: Yes

3. Have the authors made all data underlying the findings in their manuscript fully available?

Reviewer #1: Yes

Reviewer #2: No

4. Is the manuscript presented in an intelligible fashion and written in standard English?

Reviewer #1: No

Reviewer #2: Yes

5. Review Comments to the Author

Reviewer #1: All chi-square tests were performed correctly. However, the test used to compare the ages of the groups should have been the Mann-Whitney U test, not the Wilocoxon test. (The Wilxon test is applied to dependent groups).

Reviewer #2: Thanks for submitting your article for review. Here are my comments:

1. Why choose only surgical patients?

2. Was the CRP, RF evaluated in the control group?

3. The authors mention "atypical" NP migration in the title and abstract, but the term atypical is not mentioned nor explained in the manuscript. ?

4. Did the authors include definitions/thresholds for calcification and disk migration?

5. Why are you not including patient caracteristics from the control group?

6. Why do patients in the control group are included if they have a migrated disk but not the RA group?

7. How can the authors conclude that the sign is specific if they don't provide a specificity/sensibility?

8. Funding sources are not specified in the manuscript

6. PLOS authors have the option to publish the peer review history of their article (what does this mean? ). If published, this will include your full peer review and any attached files.

**Do you want your identity to be public for this peer review?** For information about this choice, including consent withdrawal, please see our Privacy Policy .

Reviewer #1: **Yes: ** Hanefi Özbek

Reviewer #2: No

---

## [Author Response · Author response to Decision Letter 1]

7 Apr 2025

Dear editors and reviewers:

Thank you very much for your careful review and constructive suggestions on our manuscript. Those comments are very beneficial for us to improve our paper. We have studied the comments carefully and tried our best to revise and improve the manuscript accordingly. We appreciate your work earnestly, and hope that the corrections will meet with approval. The answers to the reviewer’s comments are as following:

Journal Requirements:

1.Please ensure that your manuscript meets PLOS ONE's style requirements, including those for file naming. 

Answer: The article format has been modified as per your journal's requirements. The modifications to the title, format and references have all been marked in the text by inserting comments.

2.We note that your Data Availability Statement is currently as follows: “All relevant data are within the manuscript and in Supporting Information files.”

Answer: Thank you for the editor's reminder. The data included in the article is the "minimum data set". In addition, we have uploaded the original statistical data of all patients in the article as supplementary files for your review.

3.Your ethics statement should only appear in the Methods section of your manuscript. If your ethics statement is written in any section besides the Methods, please move it to the Methods section and delete it from any other section. Please ensure that your ethics statement is included in your manuscript, as the ethics statement entered into the online submission form will not be published alongside your manuscript.

Answer� The ethics statement has been removed from the acknowledgements section. The ethical statement is presented in the methods section of the original manuscript.

Reviewer 1

All chi-square tests were performed correctly. However, the test used to compare the ages of the groups should have been the Mann-Whitney U test, not the Wilocoxon test. (The Wilxon test is applied to dependent groups).

Answer: Thank you for this comment. It was our inaccuracy in the description of the methods section of the article. In fact, we used the Wilcoxon rank-sum test, which is consistent with the Mann-Whitney U test. We did not use the Wilcoxon signed-rank test. The above clarifications have been marked in the article.

Reviewer 2

1.Why choose only surgical patients?

Answer: We appreciate this insightful comment. The inclusion of surgical patients was necessitated by the requirement for high-resolution preoperative imaging (CT and MRI) and intraoperative histopathological confirmation of calcified lesions. Surgical patients offered a unique opportunity to correlate radiographic findings with direct tissue analysis, which was essential for validating the features of calcification. Furthermore, surgical candidates typically present with advanced pathology, enabling clearer visualization of radiographic signs such as nucleus pulposus migration and calcification. Although this introduces a selection bias towards more severe cases, it ensures diagnostic accuracy and facilitates direct comparisons between RA and degenerative etiologies. We have acknowledged this limitation in the revised Discussion section.

2.Was the CRP, RF evaluated in the control group?

Answer: Thank you for this comment. The control group routinely underwent CRP testing. RF was not routinely evaluated in the control group, as these biomarkers are specific to inflammatory rheumatic diseases such as RA. The control group consisted of patients with degenerative disc herniation, a condition in which systemic inflammation is not a primary feature. To clarify this point, we have added the following statement in the Methods section: "RF was assessed only in the RA group, as these markers are diagnostic criteria for RA and are irrelevant to degenerative disc disease."

3. The authors mention "atypical" NP migration in the title and abstract, but the term atypical is not mentioned nor explained in the manuscript. ?

Answer: Thank you for pointing this out. The term "atypical" refers to the diffuse calcification pattern and higher lumbar level involvement of migrated nucleus pulposus in RA patients, which contrasts with the "typical" spot-like or shell-like calcification seen in degenerative disc disease. This distinction is now explicitly defined in the Introduction and Results sections.

4.Did the authors include definitions/thresholds for calcification and disk migration?

Answer: Thank you for your insightful comment. The definitions provided in the original manuscript have been expanded for greater clarity. Specifically, calcification is now defined as hyperdense lesions exceeding 200 Hounsfield units on CT scans. Additionally, migration has been categorized as cranial or caudal displacement from the disc level as observed on sagittal MRI. These criteria are now explicitly stated in the Radiographic Assessment subsection.

5.Why are you not including patient caracteristics from the control group?

Answer: Thank you for pointing this out. The baseline characteristics of the control group (age, gender, et al) were included in Table 1. We have corrected Table 1 to display both RA and control group demographics.

6.Why do patients in the control group are included if they have a migrated disk but not the RA group?

Answer: Thank you for pointing this out. In fact, the main purpose of this study is to introduce the imaging characteristics of this atypical nucleus pulposus migration and calcification. To compare with this atypical calcification, the control group included were all patients with nucleus pulposus migration. In the comparative study, both groups were also comparing patients with nucleus pulposus migration. For instance, in the Radiographic Outcomes section, the incidence of calcification was: 5/7, 71.4% in the RA group vs 10/36, 27.8% in the control group. The direction of intervertebral disc protrusion migration: four migrated cranially and three migrated caudally in the RA group. In the control group, there were 9 patients (9/36, 25.0%) with cranial migration and 27 patients (27/36, 75.0%) with caudal migration. The above content has been marked and revised in the article.

In addition, we also observed and studied other radiological features of the RA group, such as the erosion and destruction of endplates and facet joints (described in Radiographic Outcomes). So the RA group not only included patients with nucleus pulposus migration but also encompassed all patients with rheumatoid spondylitis who underwent surgical treatment.

7.How can the authors conclude that the sign is specific if they don't provide a specificity/sensibility?

Answer: Thank you for your insightful comment. Our conclusion regarding specificity is based on the significant difference in calcification patterns observed (71.4% diffuse in RA versus 27.8% spot-like/shell-like in controls, p < 0.05). Although a sensitivity/specificity analysis would enhance our findings, the limited sample size constrains such calculations. We have tempered our conclusion to state that NPMC is suggestive of RA and recommend larger studies to validate sensitivity/specificity.

8.Funding sources are not specified in the manuscript.

Answer: Thank you for your insightful comment. The funding sources were specified following the manuscript.

Finally, we appreciate very much for these valuable suggestions, these comments are of great help for us to improve our study and hope that the correction will meet with approval. Thank you once again.

---

## [Decision Letter · Decision Letter 1]

20 May 2025

Atypical Nucleus Pulposus Migration and Calcification: A Specific Radiographic Sign for Lumbar Rheumatoid Spondylitis

PONE-D-24-49692R1

Dear Dr. Cui,

We’re pleased to inform you that your manuscript has been judged scientifically suitable for publication and will be formally accepted for publication once it meets all outstanding technical requirements.

Kind regards,

Alessandra Aldieri

Academic Editor

PLOS ONE

Additional Editor Comments (optional):

Reviewers' comments:

Reviewer's Responses to Questions

**Comments to the Author**

1. If the authors have adequately addressed your comments raised in a previous round of review and you feel that this manuscript is now acceptable for publication, you may indicate that here to bypass the “Comments to the Author” section, enter your conflict of interest statement in the “Confidential to Editor” section, and submit your "Accept" recommendation.

Reviewer #2: All comments have been addressed

2. Is the manuscript technically sound, and do the data support the conclusions?

Reviewer #2: Yes

3. Has the statistical analysis been performed appropriately and rigorously? 

Reviewer #2: Yes

4. Have the authors made all data underlying the findings in their manuscript fully available?

Reviewer #2: No

5. Is the manuscript presented in an intelligible fashion and written in standard English?

Reviewer #2: Yes

6. Review Comments to the Author

Reviewer #2: Thanks for the modifications. The corrections allows a better comprehension of the article.

7. PLOS authors have the option to publish the peer review history of their article (what does this mean? ). If published, this will include your full peer review and any attached files.

**Do you want your identity to be public for this peer review?** For information about this choice, including consent withdrawal, please see our Privacy Policy .

Reviewer #2: No

---

## [Editor Report · Acceptance letter]

PONE-D-24-49692R1

PLOS ONE

Dear Dr. Cui,

I'm pleased to inform you that your manuscript has been deemed suitable for publication in PLOS ONE. Congratulations! Your manuscript is now being handed over to our production team.

Kind regards,

on behalf of

Dr. Alessandra Aldieri

Academic Editor

PLOS ONE